# Exploring the implementation of community health worker program in Fiji: An exploratory qualitative study

Azeb Gebresilassie Tesema [1,2]*, Bindu Patel [2], Unise Vakaloloma [3], Samuel Thio [1], Devina Nand [4], Colleen Wilson [4], Rohina Joshi [1,5]

**1** School of Population Health, University of New South Wales, Sydney, Australia, **2** The George Institute for Global Health, Sydney, Australia, **3** Centre for the Prevention of Obesity and Non-Communicable Diseases, Fiji National University, Fiji, **4** Ministry of Health and Medical Services, Suva, Fiji, **5** The George Institute for Global Health, New Delhi, India

* a.tesema@unsw.edu.au

## Abstract

In Fiji, community health workers (CHWs) play a critical role in providing primary health care, especially in remote and hard-to-reach areas. This study aims to comprehensively explore the implementation of the CHW program and identify policy-implementation gaps that will strengthen the country's primary health care. We conducted a qualitative exploratory study in Fiji's six sub-divisions across the four administrative divisions—Central, Eastern, Western, and Northern. The study included six focus group discussions with 41 CHWs and six key informant interviews with CHW supervisors and nurse managers. Thematic analysis was employed to analyse the data. Participants revealed that the CHWs are engaged in range of responsibilities that improve primary health care in Fiji. The study highlighted a political commitment to strengthen the CHW program, a functional supervision process, a motivated CHW workforce, and high community trust. Despite these favourable findings and presence of a CHW policy, the study identified substantial challenges to effective implementation. These include inconsistencies in governance structure, performance variations across divisions, inadequate CHW training, irregularities in CHWs' incentives, shortage and limited capabilities of the zone nurses, as well as supply constraints affecting the Fiji's CHW program. Efforts to address these challenges within the health system should focus on strengthening the local-level governance, redefining CHW service packages, improving the quality of CHWs' pre-service and on-the-job training, creating consistent and effective incentive strategies, ensuring the provision of adequate supplies, and enhancing the data management system.

**Data availability statement:** All relevant data contributing to the findings are within the paper and in Supplementary Files.

**Funding:** The study was funded by the Australian Heart Foundation (RJ) and the University of New South Wales, through the Scientia Fellowship (RJ). The funders had no role in study design, data collection and analysis, decision to publish, or preparation of the manuscript.

**Competing interests:** The authors have declared that no competing interests exist.

## Introduction

Community health worker (CHW) programs continue to play a critical role in providing primary health care and improving population health worldwide. Their role has been crucial in reducing premature mortality through managing infectious diseases, addressing maternal and child health burdens, and preventing and managing non-communicable conditions [1–4].

Since the 1970s, Fiji has implemented a CHW program to address its health workforce shortages—falling below the recommended threshold of 2.3 health workers per 1,000 people in several areas—expand access to healthcare and improve health outcomes for its people, particularly those who reside in remote and hard-to-reach areas (e.g., maritime communities) [5–7]. As Fiji has over 300 islands and a population spread over these geographically dispersed islands, the CHW program plays a vital role in providing essential services to the population. The CHW program focuses on providing appropriate, effective, and sustainable primary health care through locally recruited and stationed CHWs, who are broadly comparable to village health workers in other settings. CHWs are community-appointed volunteers who promote health, prevent diseases, and link communities to the health system. They are selected for their commitment, integrity, and communication skills within their village. The Fiji Ministry of Health and Medical Services CHW policy defines their roles, with a primary focus on health promotion, community engagement, and health advocacy. [8–10]. As of 2020, there were around 1,600 registered CHWs in Fiji, but in the 1990s, when the CHW program was at its height, Fiji had approximately 3,000 CHWs. The significant decline over the years reflected the attention and support afforded to the program, which diminished in the last two decades [5,11].

The program was revamped in 2012 following the adoption of Fiji's Health Sector Support Program (2012–2016) and the financial support received from Australian Aid. The new initiative focused on using CHWs to provide the community's first point of contact with the health system, strengthening primary health care [8,12].

The program was further revitalized in 2015, adopting a CHW Policy that provided direction and standardised CHWs' practice. The policy called for creating a National Community Health Workers Steering Committee to act as a governance mechanism for CHWs in Fiji, including providing recommendations to the Ministry of Health and Medical Services on CHW program funding and incentive packages [5,8]. The new policy aligns with the current national development plan emphasising strengthening primary health care through recruiting, training, and retaining a qualified, motivated health workforce [6].

Despite developments in the policy front and the growing importance attached to CHW programs, previous studies have focused on specific aspects of the CHW program, such as disease-specific training, referral practices, incentives, and training needs [5,10,13,14]. Therefore, this study aimed to comprehensively explore the implementation of the CHW program, including the CHWs' recruitment, training, role assignment, supervision, and provision of supplies for the program. By examining policy, implementation, and system-level gaps, the findings from this study will

provide policymakers and stakeholders in Fiji and in the wider Pacific region a broader understanding of the CHW policy implementation and highlight the program's strengths and areas for improvement.

## Methods

### Study setting

The Fiji Islands are a republic spread over four regional divisions—Central, Western, Northern, and Eastern—further divided into 21 sub-divisions. This study was conducted in six sub-divisions selected from Fiji's four administrative divisions. The population, estimated at slightly over 900,000 people, is dispersed over 110 inhabited islands. The population is divided approximately evenly between urban and rural geographies [15–17].

Currently, the health system is based on a three-tier model that provides an integrated health service at primary, secondary and tertiary levels. However, the health programmes are divided into primary and preventive health care services (community-based) and curative health care services (hospital-based), which primarily govern the organisational structure and way of working in the health system [9]. The Ministry of Health and Medical Services Strategic Plan 2020–2025 places emphasis on reforming public health services to provide a population-based approach to diseases and the climate crisis. In recent years, the government expanded its CHW program, which outlined a role for CHWs and outreach programs for communities involving nursing stations and health centres [9]. The National Community Health Workers Steering Committee, under the Ministry of Health and Medical Services, and supported and guided by the Deputy Secretary of Public Health and the Director of Nursing Services, is responsible for the leadership and governance of Fiji's CHW program [9].

### Study design

We conducted an explorative qualitative study with CHWs, CHWs' supervisors, and sub-divisional nurse managers from 16/01/2023–27/03/2023 to gain a deeper and contextual understanding of the CHW program implementation process. We employed focus group discussions (FGDs) with CHWs and key informant interviews with nurse supervisors and divisional managers to explore the implementation of the CHW program and to identify the key enablers and barriers influencing its delivery.

### Participant recruitment and data collection

The study was co-designed with relevant stakeholders from the Ministry of Health and Medical Services. Study participants were purposively selected with the assistance of the Chief Nursing and Midwifery Officer, who communicated with nurse managers. The in-country research team organised interviews, and CHWs were contacted through their respective nurse managers. Participants with direct experience and knowledge of CHW program implementation were included in the study. FGDs included participants actively providing primary health care services in their community, while nurse supervisors and nurse managers were eligible if they supervised one or more CHWs and managed both nurses and CHWs in the community respectively. Individuals not directly involved in the CHW program or unable to provide consent were excluded. Six FGDs with 41 CHWs were conducted to explore their current roles and experiences in delivering the CHW program across six subdivisions representing Fiji's four administrative divisions. Similarly, six key informant interviews with CHW supervisors and sub-divisional nurse managers were conducted to capture their perspectives on the program and to identify health system–level enablers and barriers influencing its implementation.

Semi-structured interview guides (S1 and S2 Files) were initially developed based on the Community Health Worker Assessment and Improvement Matrix [18] and iteratively refined as interviews occurred. The interviews and FGDs with stakeholders were conducted to identify barriers and enablers for the delivery of the CHW program.

The research team (RJ, BP, and UV) conducted face-to-face interviews, and the discussion was captured using a combination of audio recording and note-taking. Permission was sought from each participant before making any recordings. The research team discussed emerging information during the data collection and points raised were considered in the subsequent interviews. This process continued until data saturation was achieved and no substantively new themes of interest were emerged [19].

## Data management and analysis

The interview audio files were then transcribed verbatim by a transcription company, Outscribe [20]. The research team checked all the transcripts independently to check for errors by simultaneously listening to the audio recording and transcripts. Data familiarisation was done by reading and making notes before data coding and then enriched by research team discussion. The transcriptions were further supplemented with notes/memos.

We coded the verbatim transcripts using NVivo 12 software (QSR International, Vic). After initial data coding by AGT, the preliminary codes and emerging themes were collaboratively reviewed and refined through reflexive team discussions to enhance the credibility and trustworthiness of the analysis. A coding framework and mind map were subsequently developed to support a rigorous analysis process. The broader Community Health Worker Assessment and Improvement Matrix (CHW AIM) informed the framework [18]. Thematic analysis was used to analyse the data. Codes and new themes were generated by iterating (abductive) between the two—inductive and deductive—approaches [21]. The draft result was then presented to national policymakers and relevant stakeholders during a high-level consultation workshop held in Fiji, where participants validate the findings and provided contextual feedback, which the research team incorporated into the final manuscript. We reported the results following the consolidated criteria for reporting qualitative research (COREQ) (S3 File) [22].

## Ethical considerations

Ethical approval was obtained from the Fiji Ministry of Health and Health Services and the University of New South Wales (UNSW) human ethics research committee. Respondents were informed about the study's aim, and all participants gave written informed consent. All invited participants agreed to participate in the study, and interviews were held where the participants were comfortable. To ensure the confidentiality and anonymity of the data, all data was de-identified before analysis, and we replaced participants' names with codes during data analysis and presentation.

## Inclusivity in global research

Additional information regarding the ethical, cultural, and scientific considerations specific to inclusivity in global research is included in the Supporting Information (S1 Checklist).

## Results

In the six FGDs, a total of 41 CHWs took part: eight from the Eastern Division (Levuka (coded in this study asFGD02)), nine from the Northern Division (Seaqaqa (FGD03)), seven from the Central Division (Nausori (FGD04) and Valelevu (FGD05)) and seventeen from the Western Division (Balevuto (FGD01) and Sigatoka (FGD06)). We conducted key informant interviews with a total of six participants —two CHW supervisors and four sub-divisional nursing managers—from different subdivisions (participants coded asKII01-KII06). All key informant participants were female, experienced sub-divisional nurses responsible for managing and supervising CHWs.

The majority of the CHWs were female, with ages ranging from 30 to 60 years. All the CHWs resided in the communities they served. Over half the CHWs had worked for more than six years (54%), followed by two to six years (24%) and those with less than two years of experience (22%). Most participants (68%) had completed secondary education, followed by tertiary education (24%) and primary education (5%) and one participant had postgraduate education. Detailed demographic and professional characteristics of the FGD participants are provided in S1 Appendix. The key results from this study are organized into eight thematic areas.

**Theme 1: Local-level leadership and governance for CHW program**

The Fiji CHW policy enables decision-making at the local level, empowering the decision-makers, such as the zone nurse, to guide the CHWs in their areas. We found a variation in the CHW program performance between divisions and sub-divisions. In some areas, particularly in the Central Division, participants reported that CHWs had received more training and were more actively engaged in community activities compared with other divisions. Although CHWs receive technical support from the zone nurse, they are accountable to the village headman (*Turaga-ni-Koro*). The headman is the community leader and supports the CHWs in their activities. We noted between-area variations in the support and involvement of the village headman. For example, some CHW participants received sub-optimal support from the village headman. CHWs explained that having personal connections (such as being related to the village headman) helped in their role and getting things done. The support that the CHWs get from the headman also depends on the headman's capacity. A CHW participant (FGD05) indicated that:

> I am looking after the village, and I do not receive any support from the headman because he's still new. Sometimes we double as an assistant for the headman and coach him what to do…

CHWs are seen as the health leads at the village-committee. Additionally, most villages and settlements have a community health committee or an advisory committee comprising community leaders and the CHWs. A CHW supervisor from the Eastern division (KII02) highlighted that *'It is always a must to have the committee in the village with the CHWs serving as heads and collaborating with the village spokesperson.'* The committee organises community meetings, advocates for health interventions, and supports community members accessing healthcare. With the CHWs, the committee facilitates reaching hard-to-reach groups, convincing community members with doubts, and engaging in various community projects, including sanitation initiatives.

**Theme 2: Recruitment of CHWs**

Most CHWs were nominated by their community during village meetings in collaboration with the zone nurses according to the selection criteria (residence and familiarity with the community, education, and ability to speak English and/or Itaukei). We observed variations in the selection process. For example, a CHW from the Eastern division (FGD02) said, *'I never volunteered. I had no option. I was the only free woman in the village. The headman asked me to be CHW'*. After the nomination, CHWs are asked to provide documents such as birth certificate, bank statement, photo ID, and more recently, evidence of COVID vaccination. Although the Ministry of I-Taukei Affairs is responsible for the recruitment of the CHWs, the Ministry of Health and Medical Services shares a letter of understanding and a document describing the CHWs' roles along with a three-year contract with the CHWs. Participants also shared that they were unaware of regulations regarding renewals or termination of CHWs' contracts, as it mainly depended on the feedback from the CHW's supervisor. *'We don't have something in black and white, but the recommendation of the nurse manager can be considered for terminating a contract,'* a nurse manager from the Central division (KII04) reported.

**Theme 3: Training of community health workers**

Although the CHW strategy requires CHWs to be trained for six weeks before deployment, we found gaps and variable implementation, depending on location and timing of deployment. CHWs recruited nearly two decades ago described completing a six-week training program that covered topics such as first aid, wound care, core competencies, safe motherhood, and child health, and concluded with formal certification. *'In 2006, I did my village Health Worker Course for six weeks…'*, CHW from the Central division (FGD04) explained. However, the newly enrolled CHWs, particularly those who started during COVID-19, didn't receive the required level of training. One FGD participant from Eastern Division (FGD02) highlighted that *'we didn't get the training [the six-week training]; even among the eight of us here, only three got just one week training.'*

CHWs and nurse managers noted that although the burden of non-communicable diseases (NCDs) is high in the community, NCD topics were minimal during the training, primarily focusing on first aid, maternal and child health programs, and palliative care. '*How do I look after an NCD case [if I don't have the training]? there's plenty of NCD cases*'(FDG01). CHWs further highlighted that they would feel confident in providing services, including for patients with NCDs if they received formal training.

Although some CHWs attended in-service refresher training, the availability, delivery, frequency, and quality of refresher training given to the CHWs varied across the sub-divisions. For example, while nurse managers are responsible for the initial and subsequent refresher training for CHWs, a nurse manager from the Central division (KII04) explained that '*To be honest, in my three years here, I didn't provide any training, just refresher during CHWs' meetings'.* Despite the challenges, participants also shared opportunities to improve the situation. '*When we want to do training, we must propose for that training as the support from the ministry is quite good'.* A nurse manager from the Western division (KII06) shared.

### Theme 4: CHW's role in the delivery of primary health care services

The Fiji CHW policy delineates health promotion and health advocacy for CHWs, which is implemented well by CHWs with the support of zone nurses. However, CHWs performed various tasks not included in the policy, which vary between sub-divisions. This is primarily due to their responsiveness to community needs, variation in resource allocation, and piloting of programs in some sub-divisions. A nurse manager from Central Division (KII05) described,

> The policy says they [CHWs] are only to do awareness or health promotion and advocacy activities. But in practice, there is a need for them also to help the nurses in taking vitals from the area. That's what we want, but the policy says otherwise.

CHWs engaged in various health related activities, including health promotion, awareness creation, sanitation, and hygiene in their villages. They maintained community profile, recorded cases, conducted home, and supported pregnant mothers. Additionally, CHWs provided primary care, offered services such as first aid, assisted patients during hospital visits, medicine refills, and linking patients with zone nurses or doctors for assistance. Although the CHWs are involved in various activities beyond what is designated in the policy, they are not allowed to perform some activities, such as measuring blood glucose or blood pressure, and doing dressings. These restrictions were not regulated consistently, and our results showed variations between divisions and sub-divisions. In areas such as the Central division, CHWs were allowed to measure blood pressure but not blood glucose. The CHWs' roles were more limited in the Northern division, as explained by a CHW supervisor(KII03);

> Before they [CHWs] were supplied with dressing materials, Panadol, and medications. But the new guidelines now are not supposed to give medication or even dress for the injuries. The only thing they can do is refer patients to the health centre.

The study participants also shared the conflict between abiding by the policy and the community's demand for service. '*… they [the community] normally ask me to perform dressing, I tell them I'm not allowed…',* a CHW (FGD05) shared their experience. In addition, lack of transportation, inability to access in hard-to-reach settlements and lack of clarity on the role, and resource constraints limited the capacity of CHWs to provide services. '*I manage three settlements. I walk a long distance. But on rainy days, I can't visit those places sometimes up to six months'.* A CHW from Northern Division (FGD03) reported.

Despite the challenges, participants discussed the significant contribution of CHWs, particularly in reducing the workload of zone nurses and as the first point of contact between the community and the health system. A nurse manager from the Western division (KII06) explained that '*All-in-all they [CHWs] are doing a good job. They are our eyes and ears out there in the community'.*

**Information management system**

We found that CHWs played a crucial role in health data management. They are involved in community profiling, recording, and reporting cases to the zone nurse every month. As participants indicated, the CHWs use booklets supplied by the Ministry to record and report their activities. *'When we go for home visits, we take our outpatient books. We record what we have done and then put it in our report books'.* A CHW from the Central division (FGD04) noted. However, as participants indicated, some CHWs used their stationery and books to record their daily activities. Furthermore, participants revealed multiple reporting formats as a challenge for the data management process. Reporting activities in multiple platforms, coupled with a paper-based system, was viewed as a challenge by the CHWs. CHWs from the Northern division (FGD03) highlighted that *'We submitted our report to the health centre, and also to the Red Cross every month'.* The unavailability of a secure place to keep documents was identified as an issue for data security. Some CHWs kept their records in the community dispensary. However, most CHW participants kept files in their houses. A CHW from the Eastern division (FGD02) shared, 'We *have children in the house. They play around with our report books'.*

**Theme 5: CHW's supervision and linkage with the health system**

We found that the CHWs' supervision process was structurally functional. Zone nurses are the immediate supervisors for CHWs, who work with the nurse manager and the medical officer. The zone nurses are a link between the CHWs and the health system and serve as the first point of contact for the CHWs. The capabilities of the zone nurse were noted to be vital to empower the CHWs and provide quality services to the community. CHWs who had good relationships with the zone nurses felt supported and empowered. A CHW from the Central division (FGD05) noted, *'…they [zone nurses] call us to come and have a meeting with them.* FGD participants from the Eastern Division also acknowledged the support provided by zone nurses. One CHW (FGD06) noted, *'They are very supportive, the zone nurses… whatever we go through in the village, we just call them and ask.'* Similarly, participants mentioned that nurse managers' support to zone nurses was critical for the CHWs to be more supported. One nurse manager (KII01) shared their experience supporting the argument: *'…as long as I'm here, I will support them [CHWs and zone nurses]'.*

**Barries for the CHW's supervision**

Despite the availability of support, the participants indicated irregularities in the supervision system. In most areas, the zone nurses made monthly supervisory visits, though the visits were supposed to be more frequent. A nurse manager from Central Division (KII05) said, *'If it were not for short of zonal nurses, they need to go out every week.'* A CHW participant from the Western division (FDG-06*) indicated that 'yes, the zone nurse comes once or twice a month'.* On the contrary, the supervision interval in some divisions was longer than a month. Notably, participants from the Eastern division received visits by zone nurses only once every three months. When this was the case, transportation problems associated with geographic and weather conditions were identified as a significant barrier to supervision.

CHWs communicate with each other and their supervisors through phone calls, Viber, and Facebook Messenger. Participants from all subdivisions shared their experience that having such a communication modality significantly helps them to receive the required support despite the geographic challenges. A CHW participant from the Eastern division (FDG-07) shared their experience; *'We usually have a chat group, and we usually put all our issues in that group'.* A nurse manager from the Central division (KII04) described a similar experience,

> They are given the contacts of the zone nurses. That is their first point of contact. If the zone nurse is unavailable, the area medical officer is looking after that area; they're also given their contacts. We [subdivision managers] also provide our contacts.

Of the challenges and proposed solutions, the CHW participants emphasized the importance of regular visits by higher-level bodies, particularly the sub-divisional nurse managers and medical officers.

### Theme 6: Incentive: implication for community health workers' motivation and program sustainability

CHWs are incentivised on their performance, linked to their monthly activity reports. The zone nurse checks the reports and submits them to the nurse manager and finally to the headquarters. Once this process is complete, the CHW is paid a $200 incentive. However, the lack of clarity about the purpose of the incentive was identified as a challenge. The policy document shared by the nurse managers stated that the allowance was meant to reimburse CHWs for the out-of-pocket costs (e.g., stationary and first aid kits) they needed for their work. In contrast, most CHW participants assumed the payment was a salary. Some of the nurse managers also had a similar understanding as a nurse manager from the Central division (KII04) explained below:

> I would say that they [CHWs] can use it to buy personal staff for their family. Also, they can use it to pay for transportation to bring their report at the end of the month. And maybe the payment can help them buy supplies for some of the programs they want to run in the community.

### CHWs use their resources to support their work

Irregular payment of incentives was identified by almost all the participants as a critical challenge for service delivery. '*The problem is that we are not receiving the payment regularly. It is not good to wait for another six or seven months'.* A CHW from the Northern division (FGD03) stated. A nurse manager from the Western division (KII01) agreed with the observations made by CHWs: *'We need to pay the CHWs on time. Sometimes, there is a backlog; they don't receive their pay for three or four months, which makes them sad'.* However, a CHW supervisor from the Northen division (KII03) disagreed, '*… it's tough to follow up with the CHWs. Most don't submit their report monthly, which delays the payment'.* Due to the delay in payments, CHWs are out of pocket for several months impacting their motivation. *'Some of the patients have no money. We spend our money and hire a taxi to transport them to the hospital.'*(CHW Northern Division: FDG03). A common theme from all the FGDs was a request to increase the incentive (ranging from $300-$600/month) and to pay that regularly.

### Community health workers' motivation to provide health services

Despite the challenges experienced, CHWs were passionate and determined to serve their community. Almost all the CHW participants enthusiastically shared their sense of service, reflecting a strong sense of responsibility toward their community and a high motivation to support and care for their people. A CHW from the Central division (FGD04) concurred:

> In my experience, it's not about money. It's about doing what I love for my community and for the people I love. Being a CHW, you are representing somebody, a Ministry of Health.

### Theme 7: Equipment and supplies for CHWs

Fiji's CHW policy does not allow CHWs to be supplied with many essential medicines and equipment, such as oral rehydration solution (ORS) or glucometers. However, the lack of clarity regarding CHWs' role and demand from the community has led to variation in the implementation of the policy and provision of supplies across different areas. The policy also notes that there is no formal documentation for the supply of medications to CHWs, and practices largely depend on each

division or sub-division. In our study, some CHWs reported purchasing basic supplies (e.g., gloves, dressings, paracetamol, and ORS) from their allowances, while in some cases, they receive them from hospitals, nearby health facilities, or individual staff. One of the CHWs from the Northern division (FGD03) noted:

> Sometimes, particularly during COVID time, we received plenty of tablets, such as Amoxycillin, the ORS, Panadol Elixir, Panadol paracetamol, and painkillers, from the hospital.

### Shortage and unregulated access to supplies by community health workers

Shortage of equipment affects CHWs' ability to deliver services to the community. A nurse manager from the Central division (KII05) indicated, *'Right now, they [CHWs] have nothing; they just have a register. Only when the zone nurse goes, they use what the nurse has brought."* CHWs encounter transportation challenges obtaining supplies from hospitals, affecting their ability to assist communities. Irregular incentives further impact supply availability, as delayed payments hinder CHWs from purchasing essential medical items. Although dispensaries were built by the community in some places, these dispensaries needed the required infrastructure and the CHWs stored the supplies in their houses. A nurse manager from the Central division (KII05) said *'they [CHWs] should have their dispensary, even in town, to keep their staff"* A CHW from the Western division (FGD06) explained more on this,

> We have a dispensary but need some support to furnish it. And we are unsure whether we or the community should buy the table, chair, and cupboards. "We need beds, chairs, and tables to use.

Despite the shortage of supplies and the requirement to have a few essential pieces of equipment to provide basic services by the Ministry, the study found that CHWs, particularly those from the Central and Western divisions, accessed these supplies themselves. A CHW discussant from the Central division (FGD04) said, *'Some of us who can afford just bought our pressure machine and the diabetes machine. Sometimes we got from Australia. It is not from the Ministry'.*

### Theme 8: Community involvement and trust toward the CHWs

The study found that, in most cases, the CHWs believed the community trusts them relies on their support at the village level. As participants mentioned, the community considered them the first person to contact and work with regarding their health. *A* CHW from the Central division (FDG-04) shared how the CHWs got the trust of their community.

> As soon as I came to her place, she said, "Oh, thank you. Now I know there's a CHW." After that, she said, "I am happy you are here. I have been asking my husband about a CHW in the village. Who at least can come around and see me".

However, the perceptions were not equally shared across communities. The CHWs believed their lack of qualification, inadequacy of subject matter knowledge, and community access to better care and information from the nearby health facilities negatively impacted the CHWs' acceptance in some areas. A nurse manager from the Western division (KII01) shared, *'Sometimes they [community members] overlook, and they don't value the CHWs, except a few. They don't see CHWs as an asset'.*

## Discussion

Fiji has recently made significant investments to expand its CHW program, facilitate the delivery of a wide range of services, and bridge the gap between the community and the healthcare system [5,10,13,14]. We observed a functioning supervision system, a motivated CHW workforce, and CHWs' perception that community trusts and relies on their support at the village level. CHWs are engaged in various activities and contributed to improving Fiji's primary health

care. The study also highlighted strong political commitment to strengthen the CHW program and support local-level decision-making.

However, despite the commitment of CHWs and an enabling policy environment, the findings identified considerable gaps in governance structure and variation in program performance across the country. In particular, the program's performance was adversely affected by inadequate CHW training, inconsistent incentives, shortage and limited capabilities of the zone nurses, and ongoing supply constraints. We discuss the implications of these findings below, drawing on relevant literature on CHW programs from other LMICs, including Pacific countries, and our collective experiences.

A clear governance structure for the CHW program is recognised as a critical factor for access to community-level healthcare. Evidence suggests that governing CHW programmes is a complex process as this program is located at the interface between the formal health system and communities. The main challenge lies when the employment parameters for CHWs fall outside the national public sector employment guidelines or are poorly integrated with the health system [23–25]. While the National Community Health Workers Steering Committee —under the Ministry of Health and Medical Services—is mandated to govern the CHW program and has played a critical coordinating role, the fact that the program is not fully integrated up to the lower level of the health system and the Ministry of Health and Medical Services is not fully responsible to the CHWs' recruitment may have contributed to the structural variations in CHW program implementation observed across jurisdictions [8]. Notably, the adverse effect had been more visible in the lower level of the health system, and empowering local health structures may help ensure better coordination and synergy of initiatives within districts [24,25]. This initiative also aligns with the experiences of many Pacific Islands Countries, which have either strengthened or established units and governance structure responsible for managing and developing their health workforce [26,27].

Evidence suggests that, in addition to governance, frequent and supportive supervision and a functional referral system that links the CHW with the formal health system are essential for improving the performance of the CHW program [28–30]. In our study, we found a strong linkage between the CHW program and the health system through the zone nurse. Supervision and CHW performance were often hindered by shortages and limited capacity among zone nurses, inadequate supplies, and transportation challenges, exacerbated by the country's geographic constraints. However, the use of alternative communication routes, such as mobile phones and social media platforms, to address these challenges and strengthen the link between the CHW and the health system is an encouraging experience that can be further harnessed within Fiji and extended to similar Pacific contexts. Nonetheless, these approaches may introduce risks related to data privacy and confidentiality, underscoring the need for standardized guidelines and secure communication protocols. Creating an enabling mechanism, involving a team of health professionals in the supervision process, continuously applying performance assessment and using community feedback can further strengthen the quality of supervision [31]. Despite the success, our study identified the shortage of nurses in primary health care as a challenge for the supervision process. Evidence suggests that the high migration of health workers across the Pacific countries and Australia and New Zealand continues to impact the accessibility and quality of health workers, particularly at the primary healthcare level [26].

Evidence from several LMICs—Bangladesh, Ethiopia, India, and Papua New Guinea—indicates that CHWs engage in various preventive and promotive health services and contribute to improved healthcare access and outcomes [1–4,32–34]. Fiji's CHWs undertake comparable activities; however, their involvement in NCD prevention, management, and rehabilitation was limited in some divisions, such as the Eastern and Northern parts, despite the high NCD burden. Our findings align with the 2016 Fiji report, which noted that CHWs primary engaged in wellness promotion, first aid, and disease prevention [35]. We also identified inconsistencies and limited role clarity across sub-divisions, consistent with earlier findings from Fiji [5]. As community needs and disease patterns evolve, the role of CHW must adapt, particularly given their strong community connections. Tailoring strategies through meaningful engagement with local communities and health workers is essential for contextual relevance and acceptability [2]. We recommend some level of standardisation of CHW's roles; however, given Fiji's diverse geography, the process must consider the variations between divisions and urban-rural settings [5,36].

Adequate, well-motivated, supported and equitably distributed CHWs are critical for a well-functioning CHW program [37,38]. This study found variations in the training levels of CHWs between divisions. The study also identified gaps in the refresher and on-the-job, training programs, including the availability, delivery, frequency, and quality of the training given to the CHWs. The challenges we observed are similar to findings in other contexts, and the findings reported in a recent study in Fiji, which recognised the need to train the CHWs to address health conditions such as tuberculosis and diabetes [7,14,35].

Effective CHW training is crucial for capacity, performance, and motivation [28,29]. Our findings highlighted the need to revise and standardise the training contents and delivery modality in a way tailored toward addressing the existing CHWs' capacity gaps and community needs, such as training that facilitates the prevention and promotion of the high NCD burden in the country. Drawing from the experiences of other Pacific countries, Fiji needs to focus on developing a competency-based curriculum that aligns with emerging population health needs [26,27,33]. Moreover, establishing educator capacity-building, and enhance training frequency and evaluation are critical [10,37,39]. Learning from other settings, digitalizing content and offering interactive online modules can support continuous learning in accessible formats [37,40].

Apart from training and supportive supervision, strategies to improve CHWs' motivation are essential to improve CHW performance. Having a mix of incentives, including recognition by the health system and community, are some factors that increase motivation [28,29]. Although we found that the delay in incentive payment discouraged some CHWs, almost all CHW participants in our study showed high motivation to continue working as CHWs and serve their community. However, the CHW's motivation may fade with time due to various reasons (such as the need for more money associated with the living cost). This is why the Fiji government needs to design consistent and practical strategies to incentivise the CHWs. In addition, if implemented, such a strategy must be aligned with the WHO's call: that they receive a financial package corresponding to their job demands, the number of hours they work, and their role [38]. Besides, adequately remunerated employment can improve health and well-being for CHWs from low socioeconomic backgrounds [41]. However, as our study and previous research in Fiji indicated, other non-financial incentives, such as training, career progression, and recognition, are equally essential to motivating the CHW in the country [5].

This study also highlighted shortage of supplies as a major bottleneck. Ensuring CHWs receive essential items—such as first aid kits—aligned with their roles, and strengthening the supply chain to regularly replenish these resources, is critical for program effectiveness. [30]. In addition, community dispensaries require adequate equipment to support CHWs' day-to-day activities. The study further identified the absence of a secure data management system, which may compromise data quality and limit its usefulness for decision-making. Promising digital health interventions used in other countries to strengthen community health information systems, streamline data collection and reporting could be adapted for Fiji [42]. However, contextualising these approaches to the local context and clearly defining the scope of CHWs' involvement is essential to avoid overburdening them with reporting responsibilities.

This study is not without limitations. Although the study included all divisions in the country, the recruitment of participants may be biased by the purposeful selection of the sub-divisions. The absence of community perspectives restricts insights to the provider side, which may not fully capture the experiences or needs of the population. Additionally, as the qualitative design limits the generalisability of the findings, future research should employ robust mixed methods approaches to generate more comprehensive evidence for informing CHW policy and program development.

## Conclusion

CHWs are critical in delivering primary healthcare services and bridging the gap between the community and the healthcare system. While the supervision process, motivation of the CHWs, and community trust remain solid, several steps are needed to strengthen the program and address inconsistencies in several areas, particularly regarding governance structure, variations in CHW roles, limited training availability, inconsistent CHW incentives, and supply constraints that have impacted the CHW program in the country. First, there is a need to empower local governance

and strengthen community engagement mechanisms. Second, the CHW service packages must be redefined based on the country's context and population needs. Third, the CHW training program must be reviewed to enhance quality and design effective on-the-job capacity-building strategies. Fourth, the system must reinforce its supervision and implement a consistent incentive system for CHWs. Lastly, there is a need to establish a clear supply chain and ensure secure data management systems.

## Supporting information

**S1 Appendix. Characteristics of the FGD participants.**
(DOCX)

**S1 File. Interview guide_Fiji CHW.**
(PDF)

**S2 File. Focus group discussion (FGD) guide_Fiji CHW.**
(PDF)

**S3 File. COREQ_checklist.**
(PDF)

**S1 Checklist. Inclusivity in global research.**
(DOCX)

## Acknowledgments

We would like to acknowledge the Australian Heart Foundation and UNSW for the financial support. The research team would also like to thank the Nursing Division at the Ministry of Health and Medical Services, Divisional Nurse managers and all the study participants. Particularly, we would like to extend our special thanks to Mr. Kishan Kumar, The Fiji Program Support Facility, and Mrs. Belinda Chan, CEO Fiji Cancer Society.

## Author contributions

**Conceptualization:** Bindu Patel, Rohina Joshi.

**Data curation:** Azeb Gebresilassie Tesema, Bindu Patel, Samuel Thio, Rohina Joshi.

**Formal analysis:** Azeb Gebresilassie Tesema, Samuel Thio, Rohina Joshi.

**Funding acquisition:** Rohina Joshi.

**Investigation:** Bindu Patel, Unise Vakaloloma, Rohina Joshi.

**Methodology:** Bindu Patel, Unise Vakaloloma, Rohina Joshi.

**Project administration:** Rohina Joshi.

**Software:** Azeb Gebresilassie Tesema.

**Validation:** Azeb Gebresilassie Tesema, Bindu Patel, Devina Nand, Colleen Wilson, Rohina Joshi.

**Visualization:** Azeb Gebresilassie Tesema.

**Writing – original draft:** Azeb Gebresilassie Tesema.

**Writing – review & editing:** Azeb Gebresilassie Tesema, Bindu Patel, Unise Vakaloloma, Samuel Thio, Devina Nand, Colleen Wilson, Rohina Joshi.

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
