## [Decision Letter · Decision Letter 0]

22 Oct 2025

PGPH-D-25-01870

Evaluation of community health worker program in Fiji: A qualitative exploratory study

Dear Dr. Tesema,

Thank you for submitting your manuscript to PLOS Global Public Health. After careful consideration, we feel that it has merit but does not fully meet PLOS Global Public Health’s publication criteria as it currently stands. Therefore, we invite you to submit a revised version of the manuscript that addresses the points raised during the review process.

A rebuttal letter that responds to each point raised by the editor and reviewer(s). You should upload this letter as a separate file labeled 'Response to Reviewers'.A marked-up copy of your manuscript that highlights changes made to the original version. You should upload this as a separate file labeled 'Revised Manuscript with Track Changes'.An unmarked version of your revised paper without tracked changes. You should upload this as a separate file labeled 'Manuscript'.Lastly, please clarify if data underlying the findings are fully available in line with the PLOS Data Policy. If not, please provide explanation and justification.

We look forward to receiving your revised manuscript.

Kind regards,

Gemma Lea Saravanos

Academic Editor

Journal Requirements:

Reviewers' comments:

Reviewer's Responses to Questions

**Comments to the Author**

1. Does this manuscript meet PLOS Global Public Health’s publication criteria?

Reviewer #1: Yes

Reviewer #2: Partly

2. Has the statistical analysis been performed appropriately and rigorously?

Reviewer #1: N/A

Reviewer #2: N/A

3. Have the authors made all data underlying the findings in their manuscript fully available (please refer to the Data Availability Statement at the start of the manuscript PDF file)?

Reviewer #1: No

Reviewer #2: Yes

4. Is the manuscript presented in an intelligible fashion and written in standard English?

Reviewer #1: Yes

Reviewer #2: No

Reviewer #1: Congratulations to the authors on the efforts and such a well-written article. The original data, albeit qualitative only, from Fiji are helpful to understand the issues/barries/enablers of the CHW program. The authors have also considered regional disparities and purposively selected interviewees from the four divisions of Fiji.

Here are several comments for the authors to address to further improve the manuscript:

1) Program evaluation usually has baseline assessment and evaluates how the outcomes of interest have changed due to the implementation of the program, via comparing with the baseline. Does this study have any baseline assessment? Or technically speaking, this study is not an evaluation of the CHW program, but only identifies current issues/barriers/enablers of the CHW program.

2) It seems unclear how the authors determined the sample size, and the justifications behind the selection of the key informants.

3) Were two independent researchers involved to generate themes? What did the study team do to ensure the results/themes summarised were unbiased?

4) The authors may need to add age and region of the respondent of the quotes;

5) Is there any analysis regarding how the issues/barriers/enablers identified are similar or different across regions, and what causes the difference, if any?

6) The discussion section is too long. Although the authors tried to mention every aspect based on the framework, a more structured organisation might help better highlight the key messages: eg. summary of key findings, key barriers and facilitators of the CHW program identified through the interviews, policy recommendations, strengths and limitations etc.

7) Authors need to expand the limitations. For example, this study design, i.e. qualitative interview with key informants only, has largely limited the interpretation and extrapolation of the results. This is just an explorative study, and more robust design using mixed methods is needed in the future to evaluate the CHW program.

Reviewer #2: The topic of community health workers (CHWs) in Fiji is highly relevant and timely, particularly in the context of rural healthcare delivery and ongoing health system challenges in Fiji. The manuscript covers important ground; however, there are several areas where clarity and depth could be improved to strengthen the overall contribution of your study. I provide some suggestions below, which I hope will help you strengthen your manuscript.

Introduction

• The introduction would benefit from a clearer and more explicit definition of CHWs upfront. This should include their roles, responsibilities, the population they serve, and the scope of medical care they are authorised to provide. Establishing this foundational understanding will better contextualise your findings and allow the reader to assess the alignment, or lack thereof, between CHW responsibilities and actual practice as described in your results.

• Please clarify whether there is a difference between CHWs and village health workers in the Fijian context. If so, this should be explicitly stated.

• Ensure all acronyms are spelled out at their first instance.

• You reference four previous studies assessing Fiji’s CHW program. It would strengthen your argument to articulate how your study builds upon or differs from this existing literature. What is the unique contribution of your study, and why was it necessary?

• Consider providing a brief discussion of the current health workforce challenges in Fiji in the introduction. This would help set the stage for understanding the importance and reliance on CHWs, particularly in rural and remote communities.

• Note: the official country name is "the Republic of Fiji".

Methods

• The methods section could be strengthened with more detail. For example:

o Was a theoretical framework used to inform or underpin the study design?

o How was the co-design with stakeholders operationalised in practice?

o What were the inclusion/exclusion criteria for participant selection?

o How were participants approached and recruited for the study?

o What experience or background did the research team have in conducting qualitative interviews?

• Please include a reference for the coding approach used in your qualitative analysis.

• A demographic table summarising participant characteristics would provide helpful context.

Results

• Several claims throughout the findings would benefit from more context or supporting evidence:

o In Theme 1, how was the assessment made that CHWs in the Central Division were better trained and more engaged?

o In Theme 3, further detail is needed on the nature of the training received by newly enrolled CHWs if it does not meet requirements.

o In Theme 5, you state that CHWs with strong relationships with zone nurses felt empowered, but the accompanying quote lacks sufficient context to support this. Consider expanding on the quote or including an additional quote that illustrates the sense of empowerment more clearly.

o Clarify what is meant by “regular visits by higher-level bodies” in Theme 5—are these supervisors, Ministry of Health officials, or others?

o In Theme 6, the phrase “sense of service” is vague and may benefit from rephrasing or further elaboration.

o In Theme 7, specify what basic supplies CHWs are purchasing out-of-pocket and what they are officially meant to be provided with. Also, the quote from Participant 03 suggests CHWs distribute a range of medications. Earlier in the manuscript, it was implied that CHWs are not authorised to dispense medication. This discrepancy requires clarification or commentary.

• Aim for consistency in phrasing throughout e.g., use either “village headman” or “village heads” consistently.

• Ensure all quotes are attributed to specific participants using participant numbers consistently.

• You may wish to reflect on the use of social media by CHWs for communicating with supervisors, and whether this presents any risks related to data privacy or confidentiality.

Discussion

• In the first paragraph, you state that "we observed……a strong community trust in the CHW program.” This claim is not clearly substantiated in the results section and should either be supported with data or rephrased.

• Consider softening definitive statements such as “the Ministry needs to empower…” to more measured language like “empowering the local level of the health system may support…”

• The limitations section needs further development to clearly acknowledge the study’s methodological constraints.

**Do you want your identity to be public for this peer review?** For information about this choice, including consent withdrawal, please see our Privacy Policy

Reviewer #1: No

Reviewer #2: No

---

## [Editor Report · Decision Letter 1]

1 Dec 2025

PGPH-D-25-01870R1

Exploring the implementation of community health worker program in Fiji: An exploratory qualitative study

Dear Dr. Tesema,

Thank you for submitting your revisions to PLOS Global Public Health for consideration.

I would like to acknowledge your careful review and adjustment of the manuscript in response to peer reviewers’ comments.

There are a few further revisions needed before we can consider the manuscript for publication and these are as follows:

**Methods:**

**1. Page 5, line 22** you state that “Six FGDs with 41 CHW and the same number of key informant interviews with divisional nurse supervisors and divisional managers were conducted across six subdivisions representing Fiji’s four administrative divisions"

It is not clear to me how many key informant interviews were conducted and with whom and why. Please explicitly state how many key informant interviews were conducted, who was interviewed and how this informed your results. You may need to describe this in a separate sentence so that your your methods are clear. This detail (in brief) should also be included in the abstract.

**Results:**

**2. Page7, line 13** “mangers” should read “managers”

**3. Page 8, line 4** “headmens” should read “headmans”

**Discussion:**

**4. Page 18, line 19** you state that the process must consider variations in geographical divisions as well as the impact of COVID-19 on the health system, however the impact of COVID-19 has not been discussed elsewhere in your manuscript, and it is not clear how this is relevant to your study. Please clearly describe the specific impact of the COVID-19 pandemic and how it is relevant to the standardisation of CHW’s roles, and why this must be considered. Else consider removing this from your discussion.

**Appendix 1:**

**5** . Thank you for providing this detailed description of FGD participants. Please consider if age as you have provided presents any privacy concerns, and report age brackets if necessary.

We look forward to receiving your revised manuscript.

Kind regards,

Gemma Lea Saravanos

Academic Editor
---

## [Editor Report · Decision Letter 2]

7 Dec 2025

Exploring the implementation of community health worker program in Fiji: An exploratory qualitative study

PGPH-D-25-01870R2

Dear Dr. Tesema,

We are pleased to inform you that your manuscript 'Exploring the implementation of community health worker program in Fiji: An exploratory qualitative study' has been provisionally accepted for publication in PLOS Global Public Health.

Best regards,

Gemma Lea Saravanos

Academic Editor